# Genetic ablation of *Immt* induces a lethal disruption of the MICOS complex

Stephanie M Rockfield[1], Meghan E Turnis[1], Ricardo Rodriguez-Enriquez[1], Madhavi Bathina[1], Seng Kah Ng[1], Nathan Kurtz[2], Nathalie Becerra Mora[2], Stephane Pelletier[3], Camenzind G Robinson[2], Peter Vogel[4], Joseph T Opferman[1]

**The mitochondrial contact site and cristae organizing system (MICOS) is important for crista junction formation and for maintaining inner mitochondrial membrane architecture. A key component of the MICOS complex is MIC60, which has been well studied in yeast and cell culture models. However, only one recent study has demonstrated the embryonic lethality of losing *Immt* (the gene encoding MIC60) expression. Tamoxifen-inducible ROSA-CreER^T2–mediated deletion of *Immt* in adult mice disrupted the MICOS complex, increased mitochondria size, altered cristae morphology, and was lethal within 12 d. Pathologically, these mice displayed defective intestinal muscle function (paralytic ileus) culminating in dehydration. We also identified bone marrow (BM) hypocellularity in *Immt*-deleted mice, although BM transplants from wild-type mice did not improve survival. Altogether, this inducible mouse model demonstrates the importance of MIC60 in vivo, in both hematopoietic and non-hematopoietic tissues, and provides a valuable resource for future mechanistic investigations into the MICOS complex.**

## Introduction

Mitochondria are responsible for many aspects of cellular biology, and the structure, organization, and dynamics of these organelles are essential for their function (1). The regions of the mitochondria are categorized as the outer mitochondrial membrane (OMM), the inner mitochondrial membrane (IMM), the inner membrane space (IMS, the region between the OMM and IMM), and the matrix (2). The IMM folds into cristae, of which the opening is known as a crista junction (CJ) (2, 3). These CJs house the mitochondrial contact site and cristae organizing system (MICOS), a large multi-protein complex (4). The MICOS complex helps form the distinctive

morphology of the mitochondrial cristae while also reaching across the IMS to connect with the sorting and assembly machinery (SAM) complex, building the mitochondrial intermembrane space bridge (MIB) (4, 5). The SAM complex resides within the OMM, where it coordinates with the transfer of the outer membrane (TOM) complex to translocate proteins into the mitochondria and to integrate β-barrel proteins within the outer membrane (6).

The MICOS/MIB complex is comprised of multiple subcomplexes that independently organize (5, 7). MIC60 (encoded by the gene *Immt*) is the largest member of the MICOS complex, and its presence at CJs stabilizes the expression and organization of other MICOS family members (5, 8, 9, 10). Many studies have detailed the disrupted cristae morphology and reduced mitochondrial respiration after MIC60 expression loss using yeast models or mammalian cell culture models via siRNA-mediated knockdown (8, 9, 11, 12, 13, 14). More recently, structural work has characterized the close interactions between MIC60 subdomains, MIC19, and SAM50, while also assessing their organization at CJs in yeast (15). However, in vivo research on the function of *Immt* in mammals has been limited. One study revealed that the loss of *Immt* is embryonically lethal in mice, with hemizygous germline deletion being detrimental to cardiac function under stress conditions (16).

To explore the role of MIC60 in adult mice, we generated a flox-*Immt* conditional knockout mouse model using the broadly expressed, tamoxifen-inducible ROSA-CreER^T2 deletion model. Relative to control mice, tamoxifen-induced *Immt* deletion reduced the expression of the MIC60 protein and of other MICOS family members and resulted in enlarged mitochondria with disorganized cristae morphology that were observed in affected tissues. Strikingly, *Immt* deletion in vivo rapidly reduced survival. Though we also identified BM hypocellularity in *Immt*-deleted mice relative to controls, BM transplant did not improve survival. Together, these results highlight the importance of MIC60 function on MICOS complex expression and mitochondrial morphology in vivo. With

[1]Department of Cell and Molecular Biology, St. Jude Children's Research Hospital, Memphis, TN, USA [2]Electron Microscopy, Department of Cellular Imaging Shared Resources, St. Jude Children's Research Hospital, Memphis, TN, USA [3]Transgenic Core Facility, Department of Immunology, St. Jude Children's Research Hospital, Memphis, TN, USA [4]Comparative Pathology Core, Pathology Department, St. Jude Children's Research Hospital, Memphis, TN, USA

Correspondence: joseph.opferman@stjude.org
Ricardo Rodriguez-Enriquez's present address is the Paul F. Glenn Center for Biology of Aging Research, Salk Institute for Biological Studies, La Jolla, CA, USA
Stephane Pelletier's present address is the Indiana University Genome Editing Center, Indiana University School of Medicine, Indianapolis, IN, USA

this research, we present a conditional mouse model system that will be beneficial for future, more targeted, research into *Immt* function and regulation.

# Results

### *Immt* deletion is lethal

We assessed the protein expression of two MICOS family members, MIC60 and MIC10, across multiple tissues from adult mice. As shown in Fig 1A, MIC60 and MIC10 protein expression was the highest in mitochondria-rich tissues such as the heart, liver, kidney, and skeletal muscle (17). We also observed the presence of multiple MIC60 isoforms, which are reportedly generated through alternative splicing, especially in the kidney (18). Because constitutive deletion of *Immt* is embryonically lethal (16), we generated a conditional knockout mouse line for *Immt* by introducing intronic loxP sites on either side of exon 3, which is present in all murine *Immt* isoforms. Cre recombinase targeting of these loxP sites would retain the first 40 amino acids from exons 1 and 2 but would cause a frameshift coding for five alternate amino acids before resulting in a premature stop codon ahead of the predicted transmembrane domain of MIC60 (Fig 1B). These $Immt^{F/F}$ mice were then crossed with ROSA-CreER$^{T2}$ mice, for tamoxifen-induced wide expression of Cre recombinase across many tissues (Fig 1B) (19). The resulting offspring were then intercrossed until we obtained $Immt^{WT}$ (hereafter referred to as WT), $Immt^{F/WT}$, and $Immt^{F/F}$, all expressing two copies of ROSA-CreER$^{T2}$.

We treated the adult mice with tamoxifen via oral gavage or intraperitoneal (IP) injection over a course of 5 d, while monitoring the mice (Fig 1B, bottom). Irrespective of the treatment method, the $Immt^{F/F}$ mice failed to survive more than 12 d after tamoxifen treatment (Figs 1C and S1A). In contrast, loss of a single *Immt* allele ($Immt^{F/WT}$) after tamoxifen treatment did not impair mouse survival (Figs 1C and S1A). When $Immt^{F/F}$ mice approached the humane endpoint, they had enlarged stomachs and fluid retention in their intestines, cecum, and colon—an effect that was not observed in WT (Fig 1D) or $Immt^{F/WT}$ (Fig S1B) mice. Intestinal bloating was observed in $Immt^{F/F}$ mice as early as 5 d after beginning treatment (Fig 1D). Histological analysis of the intestinal tract (small intestines, colon, and cecum) revealed apoptotic cells, vacuolar degeneration of small intestine enterocytes, dilated intestinal crypts containing bacteria, and submucosal inflammation in the $Immt^{F/F}$ mice (observed in five out of five mice) but not in WT mice (not observed in any of four mice, Fig 1E). These gross findings and histopathology are consistent with a diagnosis of paralytic ileus (impaired muscle contractions of the digestive tract [20]), malabsorption, and bacterial overgrowth. The combined effects of a compromised intestinal mucosa, small intestinal bacterial overgrowth, and paralytic ileus likely result in decreased water absorption and endotoxemia, which ultimately lead to severe dehydration and death.

### Tamoxifen treatment efficiently deletes *Immt* in the small intestines and affects mitochondrial morphology

Given the rapid lethality induced by tamoxifen treatment in $Immt^{F/F}$ mice, we investigated the deletion efficiency over time in multiple tissues. Our assessment focused on comparing $Immt^{F/F}$ to WT as $Immt^{F/WT}$ mice responded similarly to tamoxifen treatment as WT mice (Figs 1C and D and S1A and B) and did not show reduced MIC60 protein expression after tamoxifen treatment in any assessed tissue (Fig S1C). Genotyping of small intestines showed complete loss of the floxed *Immt* region and deletion of exon 3 as early as 3 d after beginning tamoxifen treatment (Fig 2A). In the colon and the spleen, we detected the presence of the deleted region at day 3, although the floxed alleles were not completely absent even when the mice reached an endpoint (days 7–12, Fig 2A). MIC60 protein expression was also reduced as early as day 3 of treatment in the small intestine, the colon, and the spleen (Fig 2B) but was not affected in the kidney, heart, or skeletal muscle tissues (Fig S1C). Densitometric analysis of MIC60 demonstrates that approximately half the protein is lost by day 3 in the small intestine and colon and by day 5 in the spleen (Fig S1D). Reduced MIC60 protein expression corresponded with reduced protein expression of other MICOS members (MIC10 and MIC19) without affecting the expression of another mitochondrial protein TOM20, although SAM50 expression was not clearly reduced across the assessed tissues (Fig 2B). Liver tissues showed reduced MIC60 expression when mice were moribund (days 7–12 after tamoxifen treatment), but not all MICOS/MIB family members were affected (Fig S1E).

As a core component of the MICOS complex, the loss of MIC60 expression would be expected to affect mitochondrial morphology (8, 9, 11, 12, 13, 14). Using immunohistochemical staining of COX1 (gene name *Mtco1*), a subunit of complex IV of the electron transport chain (21), we found that mitochondrial staining appeared disorganized in the small intestine, colon, and cecum but was unchanged in the liver or kidney (Figs 2C and S2A). High-resolution scanning transmission electron microscopy (STEM) images of the small intestine and colon were used to assess mitochondrial ultrastructure. As shown in Fig 2D, mitochondria from *Immt*-deleted intestines were significantly larger in area compared with WT, with aberrant cristae morphology. We also identified significantly increased mitochondrial perimeter, major axis length, minor axis length, and circularity in $Immt^{F/F}$ small intestine and colon, relative to WT tissues (Fig S2B and C). Altogether, these results demonstrate that conditional *Immt* deletion in vivo recapitulates the effects on MICOS expression and mitochondrial morphology that have been previously reported from in vitro studies (8, 9, 11, 12, 13, 14).

### Hematopoietic *Immt* deletion leads to BM failure

Aside from the abovementioned intestinal aberrations, another pathological finding after *Immt* deletion with tamoxifen was reduced cellularity within the BM that was not observed in WT mice (Fig 3A, top panels, and Fig 3B). This BM hypoplasia corresponded with reduced expression of phosphorylated histone-3 as a marker of cellular proliferation (0.64% positive cells/$\mu$m$^2$ in $Immt^{F/F}$ versus 3.9% positive cells/$\mu$m$^2$ in WT, Fig 3A, bottom panels). Genotyping confirmed that the floxed *Immt* allele was completely absent in HSPCs as early as day 3 (Fig 3C), and this corresponded with reduced MIC60 protein expression (Fig 3D). In the BM, immunohistochemical staining for COX1 showed aggregated mitochondria in tamoxifen-treated $Immt^{F/F}$ mice relative to WT mice (Fig 3E).

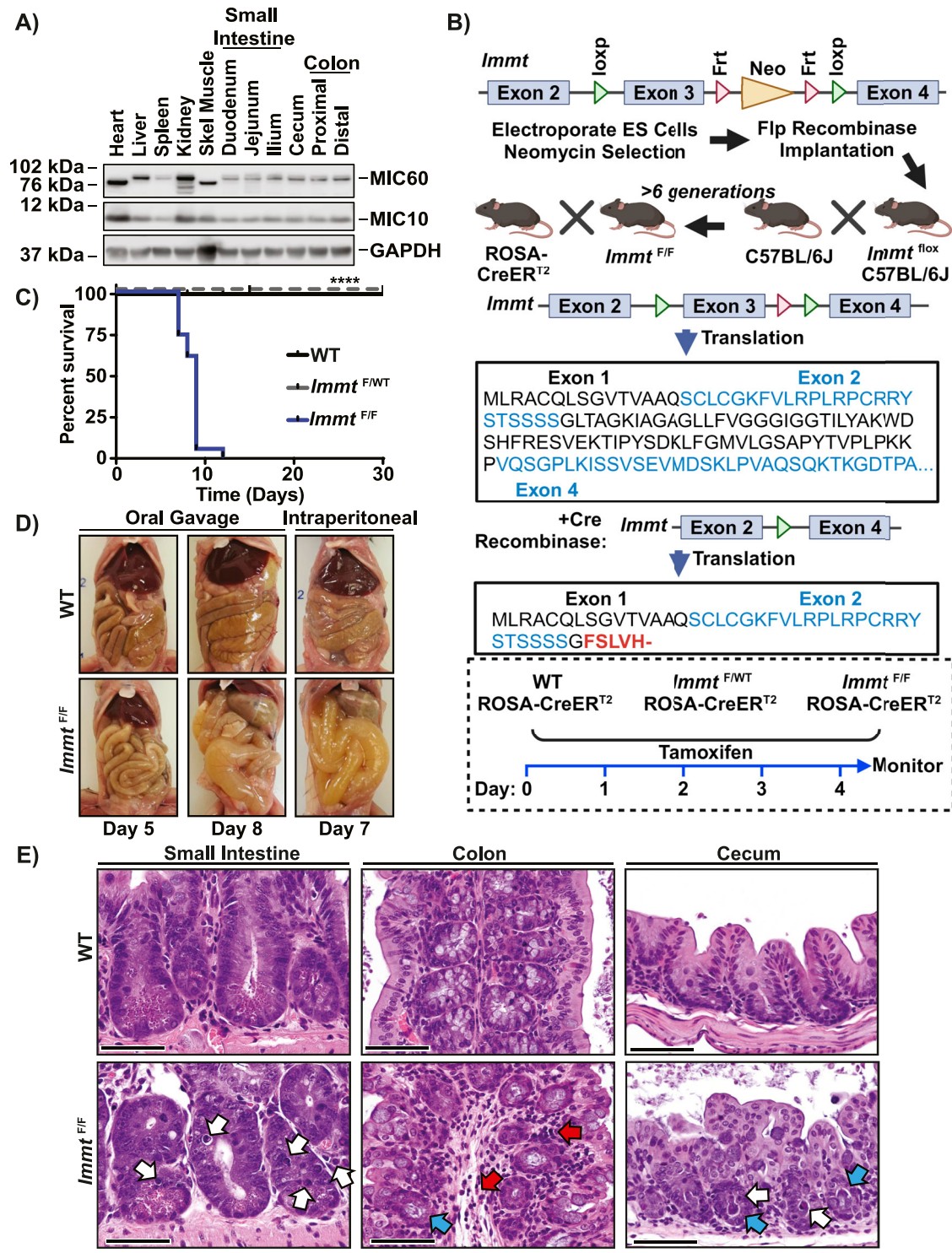

**Figure 1. *Immt* deletion is lethal.**
**(A)** Western blot demonstrating the expression of MIC60 and MIC10, relative to GAPDH, across a variety of murine tissues. Data represent three independent experiments. **(B)** Diagram depicting the placement of loxP sites to generate tamoxifen-inducible *Immt* deletion in mice, the predicted translation of MIC60 protein after excising exon 3, and the experimental procedure for tamoxifen treatments. The figure was prepared using BioRender. **(C)** Kaplan–Meier survival curve of tamoxifen treatment in WT (n = 9), *Immt*[F/WT] (n = 7), and *Immt*[F/F] (n = 23). The log-rank Mantel–Cox test was used to assess significance, P ≤ 0.0001. **(D)** Representative images of murine digestive system up to 8 d of tamoxifen treatment by oral gavage or intraperitoneal injection. **(E)** Representative images of H&E-stained small intestine (left), colon (center), or cecum (right) tissues 7 d after tamoxifen treatment. White arrows indicate apoptotic cells, red arrows indicate submucosal inflammation and edema, and blue arrows indicate dilated crypts with bacteria. Images are at 40X magnification (scale bar = 100 μm).

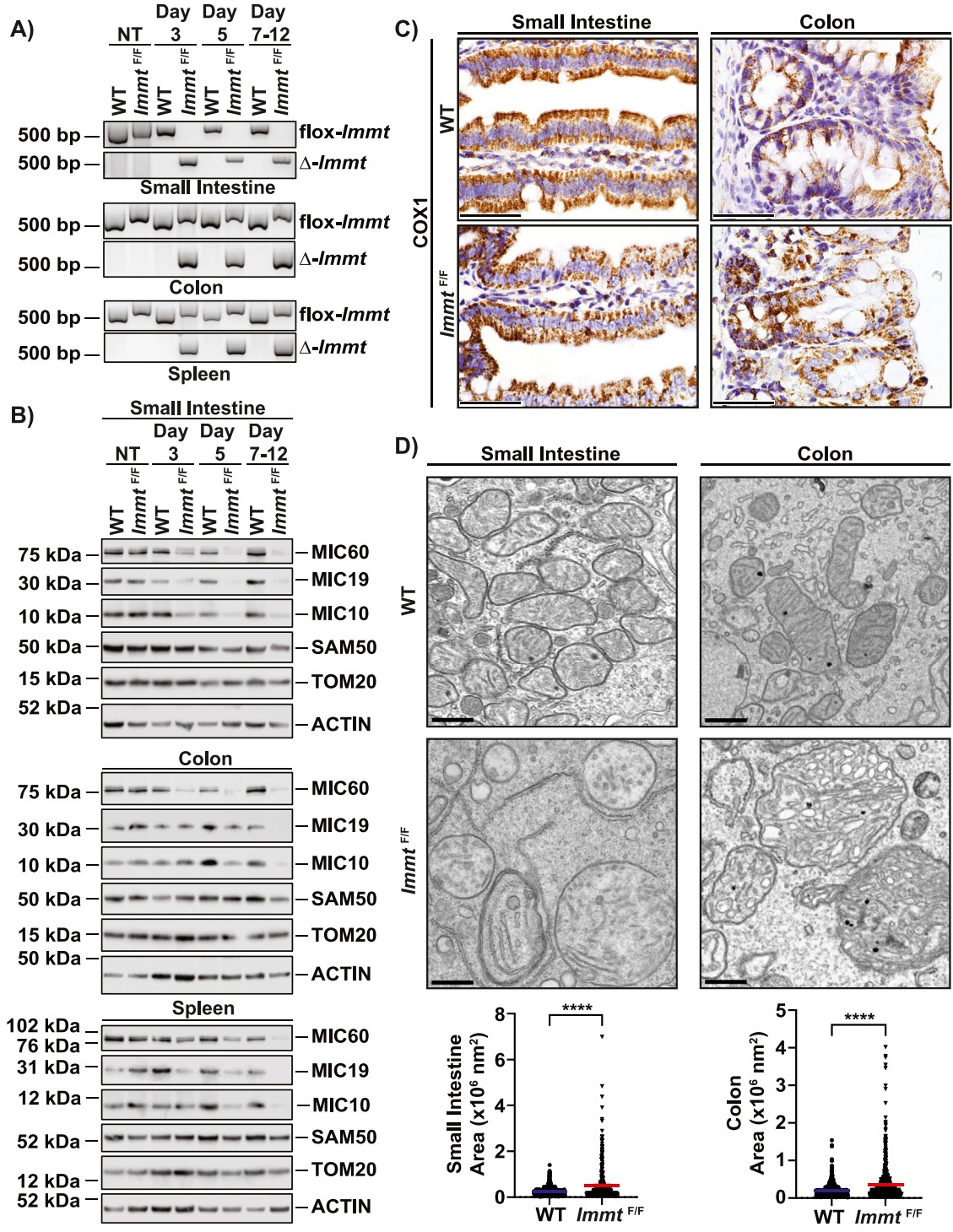

**Figure 2. Administering tamoxifen via oral gavage efficiently deletes *Immt* in the small intestine and affects mitochondrial morphology.**
**(A)** PCR results from genomic DNA isolated from the indicated tissues over time (NT = not treated). Representative samples from each timepoint are presented.
**(B)** Western blot analysis of MICOS and MIB proteins over time in the small intestine (top), colon (middle), and spleen (bottom). Representative samples from each timepoint are presented. **(C)** Representative 60x images of immunohistochemical staining for COX1 at 7 d post tamoxifen treatment in the small intestine and colon (scale bar = 50 μm). **(D)** Representative scanning transmission electron microscopy images on small intestine and colon tissues (scale bar = 500 nm). Quantification of the mitochondria area from two independent mice per condition is shown at the bottom, displaying all assessed mitochondria. For the small intestine, 747 WT and 428 *Immt*[F/F] mitochondria were measured, whereas 996 WT and 783 *Immt*[F/F] mitochondria from colon tissue were assessed. The *t* test was completed, $P \leq 0.0001$ for both tissues, and error bars represent SEM.
Source data are available for this figure.

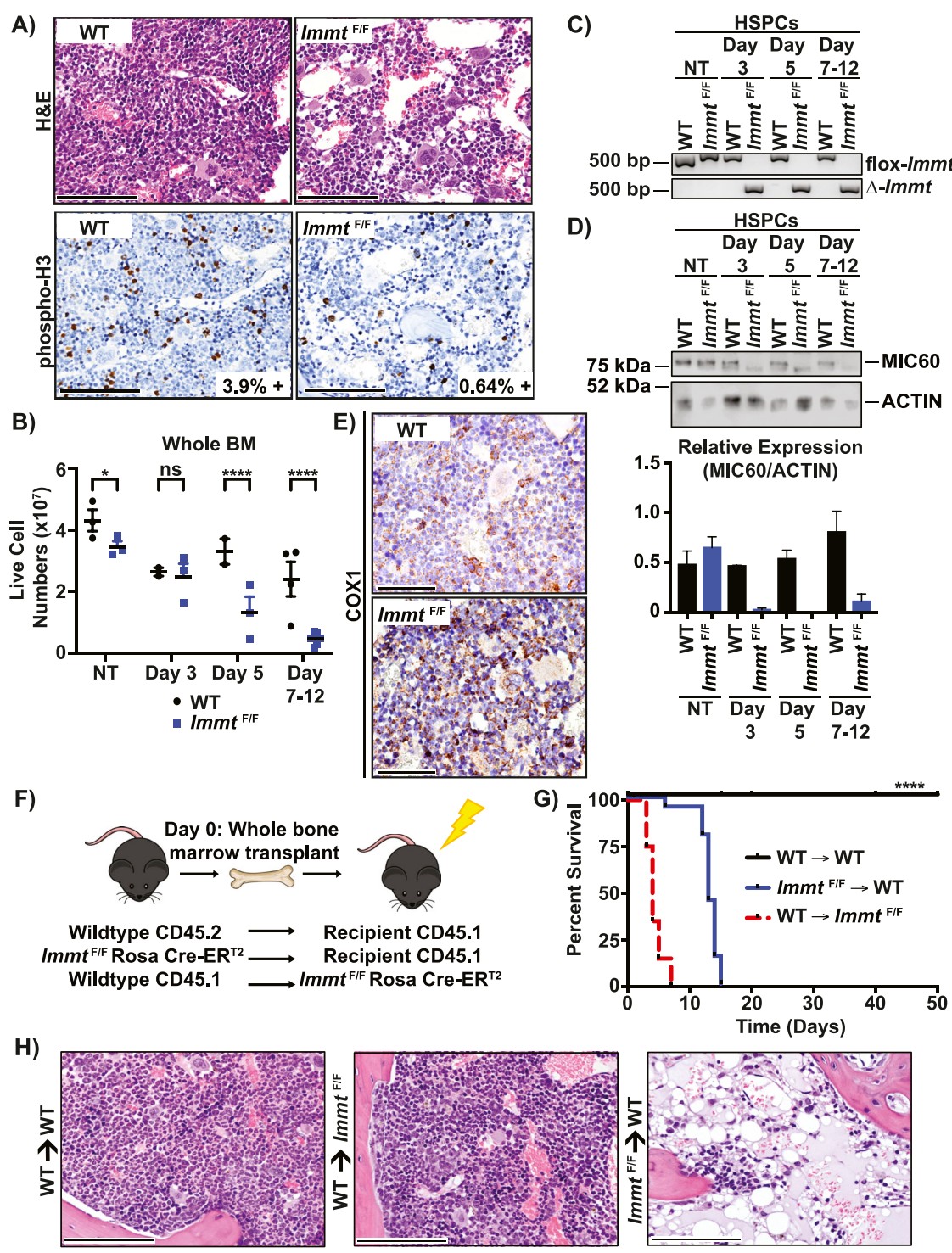

**Figure 3. Hematopoietic *Immt* deletion leads to BM failure.**
**(A)** Representative images of H&E (top) or phosphorylated histone 3 (bottom) stained BM 7 d after tamoxifen treatment. Images are at 40X magnification (scale bar = 100 μm). **(B)** Cell counts from BM after tamoxifen treatment for 7–12 d (when mice reached an endpoint). Each point reflects an individual mouse. The *t* test was completed; $P \geq 0.05$ (not significant, ns), $P \leq 0.05$ (*), $P \leq 0.01$ (**), $P \leq 0.001$ (***), and $P \leq 0.0001$ (****). Error bars represent SEM. **(C)** PCR results from genomic DNA isolated from HSPCs over time (NT, not treated). Representative samples from each timepoint are presented. **(D)** Western blot analysis of MIC60 protein over time in HSPCs. Representative samples from each timepoint are presented, and densitometric analysis for MIC60 (relative to actin) is graphed (bottom). Error bars represent SEM. **(E)** Representative 60x images of immunohistochemical staining for COX1 in the BM at 7 d post tamoxifen treatment (scale bar = 50 μm). **(F)** Diagram summarizing the experimental procedure of BM transplants. **(G)** Kaplan–Meier survival curve of tamoxifen treatment in WT (CD45.2⁺) BM transplanted into WT mice (CD45.1⁺, n = 16), WT (CD45.1⁺) BM transplanted into *Immt*^F/F mice (ROSA-CreER^T2 [CD45.2⁺], n = 20), and *Immt*^F/F (ROSA-CreER^T2 [CD45.2⁺]) BM transplanted into WT mice (CD45.1⁺, n = 21). The log-rank Mantel–Cox test was used to assess significance, $P \leq 0.0001$. **(H)** Representative 40X images of H&E-stained BM from the indicated transplant experiments, after tamoxifen treatment (scale bar = 100 μm).

The reduced BM cellularity in tamoxifen-treated *Immt*[F/F] mice suggests that inducible *Immt* deletion may trigger BM failure. To test this hypothesis, WT and *Immt*[F/F] mice were lethally irradiated and then reconstituted with donor BM from either WT or *Immt*[F/F] ROSA-CreER[T2] mice, followed by tamoxifen treatment after BM engraftment (Fig 3F). As shown in Fig 3G, transplanting *Immt*[F/F] BM into WT mice improved their survival after tamoxifen-induced deletion to 2 wk (blue line), which corresponds with the typical time course of BM failure after lethal irradiation (22). However, transplanting WT BM into *Immt*[F/F] mice failed to rescue survival; in fact, it markedly worsened their survival relative to whole-body *Immt* deletion (dotted red line, Fig 3G, compared with the blue survival curve in Fig 1C). Representative H&E images of the transplanted mice depict exacerbated hypocellularity after tamoxifen treatment of *Immt*[F/F] BM transplanted into WT mice relative to WT controls, although the WT BM transplanted into *Immt*[F/F] remained normal (Fig 3H).

We then analyzed the complete blood count (CBC) and BM populations from WT BM transplanted into WT mice and *Immt*[F/F] BM transplanted into WT mice after tamoxifen treatment (Fig 4A). Except for reticulocyte numbers and T-cell numbers, we found that all *Immt*-deleted CBC and BM parameters were significantly decreased compared with the WT controls (Fig 4A), corresponding with the significantly reduced RBC count, all of which indicates BM stress (23). All together, these results highlight the importance of functional MIC60 in BM reconstitution.

In addition, we assessed the same CBC and BM parameters when transplanting WT BM into *Immt*[F/F] mice, relative to WT controls after tamoxifen treatment (Fig 4B). Rapidly repopulating cells such as neutrophils (NE) and monocytes (MO) were observed in recipient blood within 1 wk of transplant (24). Although NE and platelet counts were unchanged in *Immt*[F/F] recipient mice relative to WT mice, we observed significantly reduced white blood cell, lymphocyte (LY), and MO counts from WT BM transplanted into *Immt*[F/F] mice (Fig 4B). We also identified significantly reduced donor cells, B cells, T cells, and myeloid cells. Intriguingly, RBC count, hematocrit percent, and reticulocyte numbers were all significantly increased relative to WT controls. These results indicate a functional WT graft on its way to complete reconstitution. However, similar to the reduced survival depicted in Fig 1C, *Immt* deletion in the WT BM–transplanted mice still resulted in rapid death in fewer than 10 d after tamoxifen administration (Fig 3G, dotted red line). Indeed, histological analysis of the intestinal tract from WT BM transplanted into *Immt*[F/F] mice revealed submucosal vacuolar degeneration of enterocytes, inflammation, and sloughing of apoptotic mucosal epithelial cells (observed in 12 out of 12 mice), but these effects were not observed in any of the WT controls (0 out of 10 mice, Fig 4C). These results indicate that even when reconstituted with WT BM, the loss of MIC60 in non-hematopoietic tissues causes lethality.

## Discussion

Herein, we characterize a conditional *Immt* mouse model; tamoxifen treatment in floxed-*Immt*, ROSA-CreER[T2]–expressing adult mice efficiently deleted *Immt* in multiple tissues (specifically, the small intestine and BM). In vivo ablation of *Immt* reduced MIC60 protein expression in the small intestine, colon, and spleen as early as 3 d posttreatment and corresponded with reduced protein expression of other MICOS proteins (namely, MIC19 and MIC10). However, complete loss of MIC60/MICOS expression was not seen until the mice became moribund between days 7 and 12. This is consistent with the long half-life (~8 d) of MIC60 protein in cardiomyocytes, human neurons, and differentiated murine myotube cells observed previously (25, 26). Importantly, detrimental effects of *Immt* deletion were observed in both hematopoietic and non-hematopoietic tissues. Because the loss of MICOS induced in tamoxifen-treated ROSA-CreER[T2] mice occurs at the same time when the mice succumb to the paralytic ileus, we suggest that other, more targeted approaches (such as Cre recombinase driven by tissue-specific promoters) will be necessary to determine the role of MIC60 in selected tissues.

Importantly, induced loss of MIC60 expression in the small intestine and colon of adult mice resulted in increased mitochondria size and disrupted cristae morphology. These observations are similar to previous in vitro observations in yeast and cell culture models (8, 9, 11, 12, 13, 14) and demonstrate that this mouse model is suitable for future in vivo mechanistic investigations into the MICOS complex. Indeed, the ability to induce *Immt* deletion in adult mice is a powerful tool, given that germline *Immt* deletion is embryonically lethal (16). Although germline *Immt* haploinsufficiency resulted in reduced MIC60 protein expression, no effects from this loss were characterized until hearts were stressed with ischemia/reperfusion (16). Our inducible system did not show an effect on MIC60 protein levels of heterozygous *Immt*[F/WT] mice in the tissues and timepoints that we assessed. Similar to a previous study, we did not observe detrimental effects from the loss of a single *Immt* allele, though we did not induce external stress in our model (16).

There are reports of *Immt* mutations in relation to human diseases. Heterozygous, missense mutations within the mitochondrial targeting sequence of *IMMT* were identified in a population of patients with either sporadic or familial Parkinson's disease (27). When these mitochondrial targeting sequence mutations were expressed in *Drosophila melanogaster*, MIC60 did not properly localize to mitochondria, and this was associated with altered mitochondrial cristae morphology, impaired larva crawling ability, and a significantly reduced lifespan. These results also demonstrate the importance of functional MIC60 in a way that relates back to *IMMT* mutations found in patients with Parkinson's disease. In addition, another study identified two consanguineous cousins presenting with developmental encephalopathy that was likely caused by a homozygous mutation in *IMMT* (28). Indeed, defects in mitochondrial function commonly manifest in relation to neurodegenerative disorders (29, 30, 31). Based on these studies, one might expect to see neurological symptoms in our tamoxifen-treated ROSA-CreER[T2] *Immt*[F/F] mice. However, the ROSA-CreER[T2] used in this model does not express in the brain (19), meaning *Immt* would not be deleted in this organ after tamoxifen treatment. Despite this limitation, this flox-*Immt* mouse could easily be crossed with other inducible Cre models, such as with Synapsin 1-Cre (32), for future investigations of MIC60 function in the brain, highlighting the potential this mouse model has to expand research into MICOS and mitochondrial function.

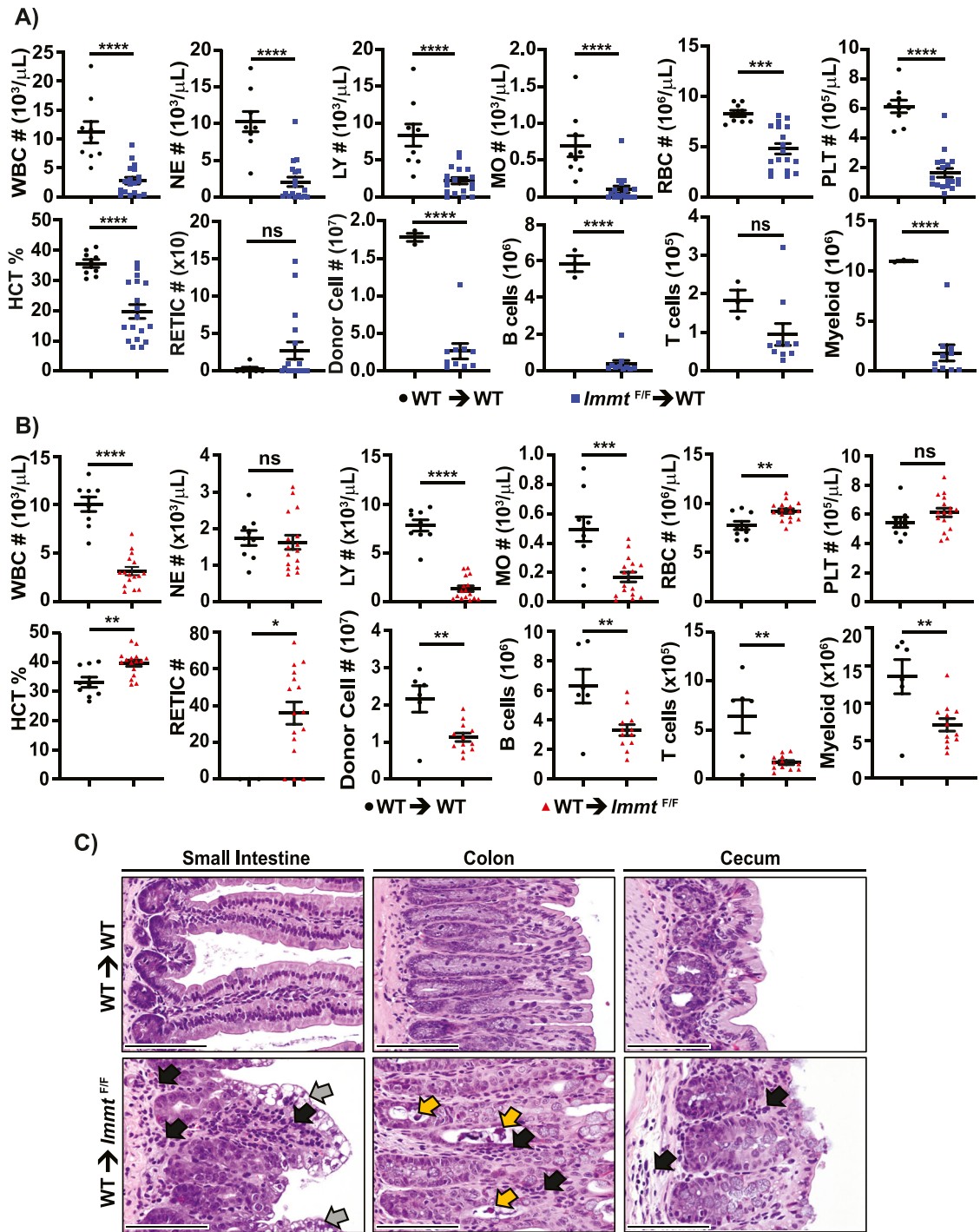

**Figure 4. WT BM transplantation does not rescue intestinal defects caused by *Immt* deletion.**
**(A)** Dot plots (where each dot represents an individual mouse) depicting complete blood count counts and donor BM cells; comparisons were made between WT BM into WT mice (black circles) and *Immt*[F/F] BM (ROSA-CreER[T2], CD45.2[+]) transplanted into WT mice (blue squares) after tamoxifen treatment. WBC, white blood cells; NE, neutrophils; LY, lymphoblasts; MO, monocytes; RBC, red blood cells; PLT, platelets; HCT%, hematocrit percent; RETIC, reticulocytes. B cells are B220+, T cells are CD3[+], and myeloid cells are CD11b+. Error bars represent SEM; the *t* test was completed; $P \geq 0.05$ (not significant, ns), $P \leq 0.05$ (*), $P \leq 0.01$ (**), $P \leq 0.001$ (***), and $P \leq 0.0001$ (****).
**(B)** Dot plots (where each dot represents an individual mouse) depicting complete blood count counts of donor BM cells; comparisons were made between WT BM into WT mice (black circles) and WT BM (CD45.1[+]) transplanted into *Immt*[F/F] mice (ROSA-CreER[T2], CD45.2[+], red triangles) after tamoxifen treatment. Error bars represent SEM.
**(C)** Representative images of H&E-stained small intestine (left), colon (center), or cecum (right) tissues after tamoxifen treatment of the indicated BM transplanted mice. Black arrows indicate inflammatory cell infiltrates, gray arrows indicate vacuolar degeneration of enterocytes, and yellow arrows indicate sloughing of apoptotic mucosal epithelial cells. Images are at 40X magnification (scale bar = 100 μm).

# Materials and Methods

### Generation of *Immt* conditional knockout mouse

*Immt* BAC clones (RP24-370K1, RP24-295O10, and RP24-268L22) were purchased from the Children's Hospital Oakland Research Institute (CHORI) BACPAC Resources. Table S1 summarizes all sequencing primers used to identify the BAC clone containing murine *Immt*, to transfer the *Immt* sequence to pBR322 (grown and selected in EL350 cells), to introduce a neomycin cassette downstream of exon 3 (surrounded by Frt flippase recognition targets), and to introduce loxP sites on the either side of exon 3 (Fig 1B; exon 3 is present in all known isoforms of *Immt*, see NCBI GeneID 76614, NC000072.7, Reference GRCm39 C57BL/6J). The constructed plasmid was electroporated into murine embryonic stem cells, then injected into early C57BL/6J mouse embryos. The embryos were implanted as previously described (33). These pups were then backcrossed to obtain *Immt*[F/F] (two alleles, with loxP sites on either side of exon 3) mice. These resulting offspring were bred with C57BL/6J mice expressing ROSA-CreER[T2] (strain #008463; Jackson Laboratories). The final genotypes of the used mice were WT, *Immt*[F/WT], or *Immt*[F/F], all with one or two copies of ROSA-CreER[T2]. Between all experiments, the age range of the mice was between 2 and 6 mo. We did not see any variation in deletion efficiency by age or by sex. Mice were kept on a 12-h light/dark cycle with food and water provided ad libitum; all procedures were approved by the St. Jude Institutional Animal Care and Use Committee (Protocol 2384).

### In vivo treatment with tamoxifen

Tamoxifen (T5648; Sigma Aldrich) was prepared in 100% sunflower oil (88921; Sigma Aldrich). 1 mg per mouse was administered via oral gavage or IP injection every day for 5 d, after which the mice were monitored for health and survival. A humane endpoint was determined based on visual assessment of hunched appearance, minimal movement, and bloated abdomen (34).

### BM transplants and isolation

Transplant studies were completed as previously described (35). WT (CD45.1[+]) or *Immt*[F/F] (ROSA-CreER[T2], CD45.2[+]) donor BM was transplanted into recipient 1,100 rad irradiated WT (CD45.1[+]) or *Immt*[F/F] (ROSA-CreER[T2], CD45.2[+]) mice such that the three experimental groups were WT (CD45.2[+]) BM into WT (CD45.1[+]) mice, *Immt*[F/F] (ROSA-CreER[T2], CD45.2[+]) BM into WT (CD45.1[+]) mice, and WT (CD45.1[+]) BM into *Immt*[F/F] (ROSA-CreER[T2], CD45.2[+]) mice. 4 wk posttransplant, ~50–70 µl of retro-orbital blood was collected for CBC analysis to verify BM engraftment. Murine health was assessed after tamoxifen treatment by oral gavage. When mice reached the humane endpoint, blood samples were again collected and analyzed via flow cytometry. After collection, RBCs were lysed before staining BM cells with the following BioLegend antibodies for flow cytometry analysis: CD3 (clone 145-2C11), CD45 (clone 30-F11), B220 (clone RA3-6B2), GR1 (clone RB6-8C5), and CD11b (clone M1/70).

To isolate HSPCs, BM was collected from both hind legs per mouse. After RBC lysis, Lin+ cells were depleted using the following BioLegend biotin-conjugated antibodies: CD3 (clone 145-2C11),

CD11b (clone M1/70), CD19 (clone MB19-1), GR-1 (clone RB6-8C5), Ter119 (clone TER-119), and B220 (clone RA3-6B2). The Lin– population was further enriched using magnetic beads conjugated to CD117/cKit (clone 2B8, Miltenyi), and HSPC enrichment was verified via flow cytometry with Sca-1 (clone D7) and CD117/cKit (clone ACK2) antibodies.

### Immunohistochemical and pathological studies

When tamoxifen-treated mice reached the defined humane endpoint, they were euthanized via $CO_2$, and tissues were collected for analysis. For immunohistochemistry analysis, tissue samples were excised, rinsed in PBS, and fixed in 4% PFA. All samples were provided to the Comparative Pathology Core (St. Jude Children's Research Hospital) for paraffin embedding, slicing, and staining with H&E, anti-phosphorylated histone 3 antibody (IHC-00061, 1:200; Bethyl Laboratories), or COX1 (ab14705, 1:4,000; Abcam). For all experiments, slides were independently analyzed and quantified by a board-certified pathologist.

### STEM

Tamoxifen-treated mice were perfused with PBS followed by electron microscopy-grade 4% PFA in 0.1 M phosphate buffer (pH 7.2). Sections of the large intestine and colon were excised and flushed with EM fixative (2.5% glutaraldehyde, 2% PFA in 0.1 M phosphate buffer) before storing in EM fixative. Samples were then postfixed at room temperature in reduced osmium tetroxide (1% osmium tetroxide plus 1.5% potassium ferrocyanide in 0.1 M cacodylate buffer) for 90 min and rinsed in ddH2O before contrasting with 4% aqueous uranyl acetate at 4°C. Dehydration was by an ascending series of ethanol to 100% followed by 100% propylene oxide. Samples were infiltrated with increasing concentrations of EmBed-812 in propylene oxide to 100% EmBed-812 and were polymerized at 60°C. Embedded samples were sectioned at ~70 nm on a Leica ultramicrotome and examined in a Zeiss Gemini 460 SEM equipped with a STEM detector. Unless otherwise indicated, all reagents were from Electron Microscopy Sciences. For quantification, TIFF images were uploaded to FIJI/ImageJ (NIH, Version 1.54 h, 15 December 2023) and assessed as described previously (36). Briefly, the perimeters of mitochondria fully represented in the field of view were manually drawn, and these were used to obtain area, perimeter, and shape descriptor measurements (including major/minor axis lengths and circularity).

### Genotyping

DNA was isolated from HSPCs using a DNeasy kit as per manufacturer's instructions (QIAGEN). Tissues were lysed overnight at 55°C in lysis buffer comprised of 100 mM Tris–HCl (pH 8), 5 mM EDTA, 0.2% SDS, 200 mM NaCl, and 1 mg/ml Proteinase K before precipitating with 2-propanol. The primer pairs used to assess the presence of the flox alleles and validate *Immt* deletion are detailed in Table S1. The PCR cycling steps were as follows: 5 min at 94°C; 30 cycles of 1 min at 94°C, 1 min at 57°C, 1 min at 72°C; and a final soak for 5 min at 72°C. The resulting products were run on 2% agarose gels containing 0.2x GelRed (Gold Biotechnology, Inc.), and bands were visualized on a ChemiDoc-MP Transilluminator (Bio-Rad).

## Immunoblotting

Tissue fragments were flash-frozen in liquid nitrogen upon collection, then broken apart with a tissue pulverizer (Bessman). Tissue powder was lysed in RIPA buffer containing protease inhibitors for 30 min on ice. Protein levels from tissues were assessed as previously described (37). For HSPCs, cells were counted, then aliquoted, pelleted, and flash-frozen. Cell pellets were lysed in 1x Laemmli Buffer with 1x protease inhibitors (Sigma-Aldrich), at a concentration of 25,000 or 50,000 cells per 10 μl. All lysates were boiled at least 5 min (95°C) before running on Bis-Tris SDS–PAGE gels and transferring to PVDF membranes. Immunoblots were horizontally cut to assess proteins of varying molecular weights and incubated overnight at 4°C in freshly prepared primary antibodies diluted in 5% BSA-PBS-T. The antibodies used were MIC60 (Bethyl Laboratories), MIC19 (Invitrogen), MIC10 (GeneTex), SAM50 (Abcam), TOM20 (Santa Cruz), actin (Millipore Sigma), and GAPDH (Cell Signaling Technologies). After washing in PBS-T and incubating in the respective anti-mouse or anti-rabbit HRP-conjugated secondary antibodies, chemiluminescence was read on a Li-COR Odyssey Fc Imager (Li-COR Biosciences) using Image Studio (Li-Cor version 5.2). No changes were made to image brightness or contrast after acquisition, and if required, blots were adjusted for horizontal alignment using ImageJ (version 1.52k, NIH). Densitometry analysis was completed using Image Lab (version 6.0.1 build 34; Bio-Rad).

## Statistical Analyses

All graphs were prepared and analyzed in GraphPad Prism (version 10.1.2[324]). For murine survival studies, a log-rank Mantel–Cox test was performed, followed by a Gehan, Breslow, and Wilcoxon test. For mitochondria size and CBC populations, a standard nonparametric $t$ test was completed. Statistical significance was set to $P \geq 0.05$ (not significant, ns), $P \leq 0.05$ (*), $P \leq 0.01$ (**), $P \leq 0.001$ (***), and $P \leq 0.0001$ (****).

# Data Availability

All data are incorporated into the article and its online supplementary material. Data for mitochondrial quantification (Figs 2D and S2B and C) are available in the online Source Data.

# Supplementary Information

# Acknowledgements

This work was supported by the American Lebanese Syrian Associated Charities (ALSAC) of St. Jude and NCI P30 CA021765 for the STEM imaging. In addition, the authors would like to thank the members of the Opferman Laboratory (St. Jude Children's Research Hospital) for helpful discussions. Finally, the authors thank the St. Jude Children's Research Hospital Animal Resource Center, Comparative Pathology Core, Electron Microscopy Imaging Core, and the Flow Cytometry and Cell Sorting Shared Resources for their support of this project.

## Author Contributions

SM Rockfield: conceptualization, data curation, formal analysis, validation, investigation, visualization, methodology, project administration, and writing—original draft, review, and editing.
ME Turnis: conceptualization, data curation, formal analysis, validation, investigation, visualization, methodology, and writing—review and editing.
R Rodriguez-Enriquez: conceptualization, data curation, and formal analysis.
M Bathina: conceptualization and data curation.
SK Ng: conceptualization, resources, and data curation.
N Kurtz: investigation.
N Becerra Mora: investigation.
S Pelletier: conceptualization and resources.
CG Robinson: supervision and investigation.
P Vogel: resources, data curation, and formal analysis.
JT Opferman: conceptualization, formal analysis, supervision, funding acquisition, project administration, and writing—original draft, review, and editing.

## Conflict of Interest Statement

The authors declare that they have no conflict of interest.

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
