## [Reviewer comments · Life Science Alliance]

Life Science Alliance

Genetic ablation of Immt induces a lethal disruption of the MICOS complex

Stephanie Rockfield, Meghan Turnis, Ricardo Rodriguez-Enriquez, Madhavi Bathina, Seng Kah Ng, Nathan Kurtz, Nathalie Becerra Mora, Stephane Pelletier, Camenzind Robinson, A. Peter Vogel, and Joseph Opferman

DOI: <https://doi.org/10.26508/lsa.202302329>

Corresponding author(s): Joseph Opferman, St. Jude Children's Research Hospital

Review Timeline:

Submission Date:	2023-08-22
Editorial Decision:	2023-09-19
Revision Received:	2024-01-03
Editorial Decision:	2024-01-23
Revision Received:	2024-02-23
Editorial Decision:	2024-02-27
Revision Received:	2024-02-29
Accepted:	2024-03-01

Transaction Report:

September 19, 2023

Re: Life Science Alliance manuscript #LSA-2023-02329-T

Dr. Joseph T Opferman
St. Jude Children's Research Hospital
Biochemistry
262 Danny Thomas Place
MS 340
Memphis, TN 38105

Dear Dr. Opferman,

Thank you for submitting your manuscript entitled "Inducible genetic ablation of Immt induces a lethal disruption of the MICOS complex" to Life Science Alliance. The manuscript was assessed by expert reviewers, whose comments are appended to this letter. We invite you to submit a revised manuscript addressing the Reviewer comments.

Thank you for this interesting contribution to Life Science Alliance. We are looking forward to receiving your revised manuscript.

Sincerely,

B. MANUSCRIPT ORGANIZATION AND FORMATTING:

Reviewer #1 (Comments to the Authors (Required)):

In this manuscript by Rockfield et al, the authors explore the effect of inducible depletion of the central MICOS complex member, MIC60, in adult mice. The authors find that, in accordance with numerous recent studies linking MICOS complex mutations with infantile lethality in humans, tamoxifen-induced MIC60 depletion caused lethality in mice within 9 days. They further characterize the pathology of the mice, finding that they have intestinal and bone marrow defects, which correlates with tissues where MIC60 protein levels are most reduced. They use cycloheximide treatment of MEFs to argue that MIC60 has a relatively long half-life, and make the case that the selective depletion of MIC60 observed in different tissues can be attributed to the protein being long-lived. While this study establishes a nice in vivo model to study MIC60 depletion in adult mice that the authors propose to use in follow up works, the findings of the study itself are somewhat limited in scope as they only explore the pathology of a subset of tissues and do not characterize or attribute pathology to mitochondrial defects. Importantly, the conclusion that differential MIC60 reduction in different tissues can be explained by protein half-life cannot be reached without further supportive evidence. As this study touts a new in vivo model to study loss of MIC60, it should also be made clear if the MIC60 deletion occurs genotypically in all the tissues examined in a relatively uniform timeframe.

Specific points:

1. The authors conclude that MIC60 protein half-life in different tissues explains the lack of protein depletion in tissues such as skeletal muscle or heart. However, is it also possible that tamoxifen induced knockout occurs at different timings or in a mosaic pattern in different cell types/tissues? The authors could use qPCR to determine that the MIC60 transcript is equally depleted in each tissue which would bolster their claim, and should also show that the knockout genotype is achieved at similar time-scales in different tissues.
2. To put their work into context, the authors should cite the works identifying numerous other MICOS mutations in human patients and comment on whether or not those studies identify intestinal or bone marrow defects, and why this may or may not be the case.
3. Depletion of MIC60 by knockout and inducible knockdown has been extensively performed in human tissue culture cell lines, all with more severe effects on MICOS protein levels than those observed here. Generally speaking, MIC19 and most other MICOS subunit levels are depleted in all such characterized lines (for example: Ott et al, Plos One, 2015; Li et al, Cell Death Diff, 2015; Stephan et al, EMBO J, 2020). Can the authors comment on this difference?
4. MIC19 western would be helpful to show in addition to MIC10 western in Fig. 1A and EV1, as MIC60 and MIC19 form a subcomplex together, and MIC60 has more uniform and profound effects on MIC19 stability in all cell types examined.
5. The conclusion that MIC60 is the primary "scaffold" for the complex is not supported by the experiments shown here, nor is there evidence provided that MIC60 is an "in vivo scaffold." Rather, it has been shown to be required for the maintenance of protein levels of other MICOS proteins. The Mic60-Mic19 subcomplex is proposed to act as a supportive structure that shapes cristae junctions as well as form the basis of the trans-intermembrane space MIB complex through interactions with Sam50 (for example, see Bock-Bierbaum et al, Science Advances, 2022). The language in the manuscript should be adjusted to reflect this and other recent works, which should also be cited.

Reviewer #2 (Comments to the Authors (Required)):

In this manuscript, Rockfield and colleagues reported a mouse model with inducible whole-body knockout of *Immt* which encodes mitochondrial protein MIC60, and discovered their overt phenotypes were premature death and bone marrow hypocellularity. This is an interesting observation but the conclusion made from the data needs to be carefully reevaluated. Also, it lacks the details in the method/technique for researchers to convincingly utilize the mouse model. Below are the points with details:

1. Unfortunately, this is not the first study to report an in vivo *Immt*-deletion mouse model. In the paper (PMID: 37107296), while hemizygotic deletion of *Immt* has no overt phenotype, comparable to the phenotype in this manuscript, homozygotic deletion was found to be embryonic lethal. Would this be the reason that the authors generated a tamoxifen-inducible model? Assuming

the authors did not know the reference, it is hard to infer the rationale for generating an inducible knockout model rather than a conventional approach to delete it from the embryo. Was it because the authors knew it was embryonic lethal? If so, it should be clearly mentioned in the introduction part.

2. The mouse breeding scheme and the genotypes used in the study are not well described in the method section. Did the authors cross the resulting chimera from the ES injection with wild-type first to ultimately obtain the *Immt-fl/fl* by interbreeding? Were then these *fl/fl* mice crossed with *ROSA-CreERT2*? If so, what is the exact genotype of control mice as denoted "*Immt-WT*"? Is it the *ROSA-CreERT2* mouse without the *loxP* cassette? A schematic figure would be greatly helpful for readers to understand as the major claim of this manuscript is the generation of the mouse model.

In regards to the genotype, as whole-body Cre expression itself has a hematologic defective phenotype in mice (PMID: 19380810, 37045870) akin to those the authors observed from the knockout mice in this manuscript, it is critical to include Cre control treated with tamoxifen to attribute the phenotype to the loss of *Immt*.

3. The age of mice at the treatment of tamoxifen should be indicated.

4. I am surprised that there is no single assay about mitochondria after deleting a mitochondrial gene. It is essential to know whether the *in vitro* MIC60 KO phenotype is recapitulated *in vivo* despite the limited scope of the claim in the manuscript. MIC60 loss-of-function results in mitochondrial fragmentation among many (PMID: 32567732). Because the tamoxifen did not delete the gene in all tissues and did not reduce the MIC60 level to half in mice with hemizygous allele in any of the tissues, the authors at least should check the morphology of mitochondria in the tissues shown in the manuscript and see whether the fragmentation and gene deletion result has a correlation.

5. The tamoxifen-induced gene editing efficiency in multiple tissues does not match the original paper that the author cited (PMID: 17456738). Thus, the authors should verify the gene deletion efficiency in multiple tissues by orthogonal methods such as PCR assay of exon 3 or southern blot. Also, please include the densitometry analyses of western blot to quantitatively indicate the MIC60 expression levels in different genotypes. Increasing the number of WT should be necessary.

6. Please provide the gene deletion efficiency in hematopoietic cells that the authors tested in the manuscript.

(Optional) 7. The knockout mouse phenotype apparently does not overlap with any of the human diseases associated with MICOS dysfunction (PMID: 31825482), let alone the neurological phenotypes that the authors mentioned in the discussion presumably due to the incomplete penetrance of tamoxifen to the brain. Could the authors elaborate some thoughts on the discrepancy between the observed phenotype in this manuscript and the human diseases?

Reviewer #3 (Comments to the Authors (Required)):

This study seeks to define the consequences of disrupting of a core MICOS subunit, MIC60 (gene name *Immt*), in an *in vivo* mouse model. The authors generated a global inducible *Immt* knockout and find that, strikingly, the animals die acutely in just over a week. They define two tissues demonstrating pathology that may contribute to this lethality: the intestine, which shows pathophysiology consistent with paralytic ileus, and the bone marrow, which shows hypocellularity across multiple cell types. For the most part, the experimental claims are supported by the data presented. However, there are some claims that, if explored in more detail, would substantially strengthen the manuscript. Additionally, I have a couple of technical concerns that should be addressed to more fully characterize the MIC60 inducible model. This is of particular importance as the authors deem this mouse model "a novel model system for future research." Finally, I believe this manuscript could be significantly strengthened by the inclusion of all experimental data within the main text (rather than having an extended figure), additional citations, and the inclusion of a bona fide discussion section in which they place their findings into a broader context. As it stands currently, the manuscript does not adequately integrate the current state of the MICOS literature, nor does it summarize the novelty of the system and findings to the reader.

Major revisions

1. The authors do a nice job of mapping MIC60 expression across tissues (Figure 1A) and confirming its loss upon tamoxifen treatment across many tissues of interest (Figure 1D, EV1). However, much of this data is included in EV1, and its inclusion in the main text would be helpful for the reader to more fully understand the system and evaluate the conclusions of the paper. For instance, it may be surprising that intestinal KO of *Immt* is what drives lethality in a global KO model, but upon examination of the data, this seems to be true because the knockout of MIC60 occurs in the intestine before (or more completely) than in other tissues such as the kidney and heart. Showing these data in the main text will help the reader to draw these conclusions, which are important for fully understanding the context of the study as well as the model.

2. The half-life data shown in Figure 1C is interesting, but it is overstated and somewhat irrelevant to the claims made in the paper. First, stating that MIC60 is a "long-lived protein in MEFs" because it does not decrease in abundance over 24 hours relative to MCL-1 is an overstatement. A recent study looking at isotopic labeling in HeLa cells found MCL-1 to be the shortest-

lived mitochondrial protein quantified with a half-life of 45 minutes (PMID: 34847359, Figure S2B). Furthermore, a recent study defining short lived proteins across 4 cultured cell lines found that fewer than 10% of proteins are 'short-lived' with a half-life under 8 hours following translational arrest (PMID: 34626566). Thus, the statement that MIC60 is 'long-lived', based on a cycloheximide time course over 24 hours is an overstatement; it has a longer half-life than MCL-1, but this is not surprising given these datapoints. Nonetheless, the experiment itself is flawed because mitochondrial proteins will almost certainly have different half lives in cultured cells versus in vivo. However, this can be addressed by the inducible nature of the Immt knockout mouse model profiled in the paper. The authors should use this mouse to map the half-life of MIC60 across the intestine and the colon to understand the protein turnover kinetics in these tissues. This will not only be valuable to understand MIC60 biology but would also be important for understanding the kinetics and onset of pathology relative to loss of the protein.

3. The protein expression of MIC60 is not shown for any resident cell types in bone marrow, and this should be shown for at least a subset of these cells. However, MIC60 deletion likely does not directly drive pathologies in all hematopoietic cells, such as red blood cells, which are diminished in Immt KO mice, but would be predicted to be unaffected by loss of MICOS due to their lack of mitochondria. The fact that the pathology persists in RBCs suggests that hematopoietic stem cells may be affected as well, which could be tested for MIC60 expression after tamoxifen treatment.

4. The authors note that tamoxifen is given by oral gavage in the animals, which induces lethality within 9 days. It is notable that intestinal cells would be amongst the first to encounter tamoxifen via this delivery method, which may contribute to the severity of pathophysiology in this cell type. Alternatively, it is possible that MIC60 has a relatively short half-life in the intestine relative to other tissue types, which is consistent with what the authors propose. The authors should test whether tamoxifen administration via a non-oral route, such as intraperitoneal injection, causes lethality on the same timescale and causes intestinal pathology. This need only be done on a small cohort of animals, and the full scope of the study need not be repeated. It should also be noted that if the results of this experiment show different phenotypes than what are detailed in the current figures, it does not invalidate the work, but rather suggests that the method in which researchers induce Immt knockout influences the pathology seen. Given the authors' assertions that this paper is characterizing "a valuable resource" and "a novel model system for future research", this is an important control for understanding the utility of this model, as well as the conclusions that can be drawn from future studies in which it may be used.

5. The lack of discussion in the manuscript detracts from its potential impact. There is very little context as to why the authors performed the study and what it might mean in the context of the MICOS field. There are no parallels made to human disease, despite the fact that mutations in IMMT have been linked to encephalopathy and optic neuropathy (PMID: 34842280). The authors should place their findings into the context of these and other important studies in the MICOS field.

Minor revisions:

1. "Inducible" and "induces" are both used in the title, which is redundant.
2. The histology in this paper would be more powerful if quantified and analyzed statistically.
3. The authors find loss of MIC60 in the spleen - are there signs of splenomegaly?
4. The authors claim that "To date, the in vivo impact of MIC60 loss has not been documented." However, a recent study generated Mifofilin (i.e., Immt) global knockout mice and this study should be cited. (PMID: 37107296).

Reviewer #1 (Comments to the Authors (Required)):

In this manuscript by Rockfield et al, the authors explore the effect of inducible depletion of the central MICOS complex member, MIC60, in adult mice. The authors find that, in accordance with numerous recent studies linking MICOS complex mutations with infantile lethality in humans, tamoxifen-induced MIC60 depletion caused lethality in mice within 9 days. They further characterize the pathology of the mice, finding that they have intestinal and bone marrow defects, which correlates with tissues where MIC60 protein levels are most reduced. They use cycloheximide treatment of MEFs to argue that MIC60 has a relatively long half-life, and make the case that the selective depletion of MIC60 observed in different tissues can be attributed to the protein being long-lived. While this study establishes a nice *in vivo* model to study MIC60 depletion in adult mice that the authors propose to use in follow up works, the findings of the study itself are somewhat limited in scope as they only explore the pathology of a subset of tissues and do not characterize or attribute pathology to mitochondrial defects. Importantly, the conclusion that differential MIC60 reduction in different tissues can be explained by protein half-life cannot be reached without further supportive evidence. As this study touts a new *in vivo* model to study loss of MIC60, it should also be made clear if the MIC60 deletion occurs genotypically in all the tissues examined in a relatively uniform timeframe.

Specific points:

1. The authors conclude that MIC60 protein half-life in different tissues explains the lack of protein depletion in tissues such as skeletal muscle or heart. However, is it also possible that tamoxifen induced knockout occurs at different timings or in a mosaic pattern in different cell types/tissues? The authors could use qPCR to determine that the MIC60 transcript is equally depleted in each tissue which would bolster their claim, and should also show that the knockout genotype is achieved at similar time-scales in different tissues.

In concordance to a similar comment from Reviewer 3 (see point #2), we completed kinetic experiments to induce *Immt* deletion with tamoxifen treatment in mice and assessed gene and protein expression over time. In revised Fig 2, we clearly demonstrate that the floxed *Immt* is completely absent in the small intestine as early as 3 days after tamoxifen treatment, while the floxed allele is still present at all time points in colon and spleen (all tissues also show a population of deleted *Immt* by day 3, indicating a heterogeneity of expression that the reviewer alluded to). Importantly, we observed a gradual reduction of MIC60 protein expression over time in these tissues. We believe these points address the reviewer's point and we have determined that additional qPCR to assess transcript levels was not required.

2. To put their work into context, the authors should cite the works identifying numerous other MICOS mutations in human patients and comment on whether or not those studies identify intestinal or bone marrow defects, and why this may or may not be the case.

We have included discussion of the *IMMT* mutations identified in patients with Parkinson's disease (Tsai *et al.* 2022, PMID: 29456190) and developmental encephalopathy (Marco-Hernández *et al.* 2021, PMID: 34842280), and referenced the indicated manuscripts. As Reviewer 3 noted, we administered tamoxifen through oral gavage (and in the revised manuscript, through intraperitoneal injection). The intestinal defects are likely seen due to the route of administration and the high cell turnover in these tissues, and this is further supported by

the incomplete deletion observed in other tissues (revised Fig 2 and Supplementary Fig S1). Below is the relevant Discussion passage to address the reviewer's comment:

“There are reports of *Immt* mutations in relation to human diseases. Heterozygous, missense mutations within the mitochondrial targeting sequence of *IMMT* were identified in a population of patients with either sporadic or familial Parkinson's disease (27). When these mitochondrial targeting sequence mutations were expressed in *Drosophila melanogaster*, MIC60 did not properly localize to mitochondria, and this was associated with altered mitochondrial cristae morphology, impaired larva crawling ability, and a significantly reduced lifespan. These results also demonstrate the importance of functional MIC60 in a way that relates back to *IMMT* mutations found in patients with Parkinson's disease. In addition, a recent case study identified two consanguineous cousins presenting with developmental encephalopathy that was likely caused by a homozygous mutation in *IMMT* (31). Indeed, defects in mitochondrial function commonly manifest in relation to neurodegenerative disorders (28, 29, 30). Based on these studies, one might expect to see neurological symptoms in our tamoxifen-treated *Immt*^{F/F} mice, were it not the case that the intestinal defects are too severe for the mice to survive. Additionally, the Rosa-Cre ERT² used in this model does not express in the brain (19), meaning *Immt* would not be well deleted in this organ following tamoxifen treatment. Despite this limitation, this flox-*Immt* mouse could easily be crossed with other inducible Cre models, such as with Synapsin 1-Cre (32), for future investigations of MIC60 function in the brain, highlighting the potential this mouse model has to expand the field of MICOS and mitochondrial function.”

3. Depletion of MIC60 by knockout and inducible knockdown has been extensively performed in human tissue culture cell lines, all with more severe effects on MICOS protein levels than those observed here. Generally speaking, MIC19 and most other MICOS subunit levels are depleted in all such characterized lines (for example: Ott et al, Plos One, 2015; Li et al, Cell Death Diff, 2015; Stephan et al, EMBO J, 2020). Can the authors comment on this difference?

We agree with the reviewer's assessment that the protein expression of other MICOS/MIB members (namely MIC19, MIC27, MIC10, and SAM50, now presented in revised Fig 2B–D and Supplementary Fig S1) are not as severely affected as the changes previously reported in HeLa cells (either with doxycycline induction [Ott et al. 2015, PMID: 25781180] or CRISPR/Cas9 [Stephan et al. 2020, PMID: 32567732]) or in MEFs (Li et al. 2015, PMID: 26250910). Given these are cell culture models, we think it is not surprising to obtain more complete and uniform loss of expression of these family members compared to an *in vivo* model, with differing response in different tissues (each of which have heterogenous populations of cells). Indeed, we have generated MEFs from this mouse line and observed reduced expression of MICOS/MIB family members similar to these referenced reports – these data will be included in a more mechanistic manuscript that will soon follow this submitted paper. To address the reviewer's point, we completed genotyping analyses from the assessed tissues to validate deletion efficiency and assessed MICOS/MIB protein expression over time in multiple tissue (these data are in revised Fig 2). We also included this as a point in our Discussion (please see the quoted response to point #2, above).

4. MIC19 western would be helpful to show in addition to MIC10 western in Fig. 1A and EV1, as MIC60 and MIC19 form a subcomplex together, and MIC60 has more uniform and profound effects on MIC19 stability in all cell types examined.

We have included MIC19 and SAM50 protein expression data in our samples accordingly, in response to this point and to strengthen our response to point #5.

5. The conclusion that MIC60 is the primary "scaffold" for the complex is not supported by the experiments shown here, nor is there evidence provided that MIC60 is an "in vivo scaffold." Rather, it has been shown to be required for the maintenance of protein levels of other MICOS proteins. The Mic60-Mic19 subcomplex is proposed to act as a supportive structure that shapes cristae junctions as well as form the basis of the trans-intermembrane space MIB complex through interactions with Sam50 (for example, see Bock-Bierbaum et al, Science Advances, 2022). The language in the manuscript should be adjusted to reflect this and other recent works, which should also be cited.

As mentioned in point #4 above, we now include western blotting for MIC19 and SAM50 expression in our samples to better address this point. Additionally, we removed all instances where MIC60 was referred to as a scaffold protein, and made sure to include the below references in the Introduction:

- 1) For evidence of MIC60 loss effecting the protein expression of other MICOS family members, and altering cristae morphology – Ott *et al.* 2015, PMID: 25781180; Stephan *et al.* 2020, PMID: 32567732; and Li *et al.* 2015, PMID: 26250910
- 2) Organization of the MICOS/MIB complex: Huynen *et al.* 2015, PMID: 26477565.
- 3) Organization and interactions between MIC60, MIC19, and SAM50 related to CJs: Bock-Bierbaum *et al.* 2022, PMID: 36044574

Reviewer #2 (Comments to the Authors (Required)):

In this manuscript, Rockfield and colleagues reported a mouse model with inducible whole-body knockout of *Immt* which encodes mitochondrial protein MIC60, and discovered their overt phenotypes were premature death and bone marrow hypocellularity. This is an interesting observation but the conclusion made from the data needs to be carefully reevaluated. Also, it lacks the details in the method/technique for researchers to convincingly utilize the mouse model. Below are the points with details:

1. Unfortunately, this is not the first study to report an in vivo *Immt*-deletion mouse model. In the paper (PMID: 37107296), while hemizygotic deletion of *Immt* has no overt phenotype, comparable to the phenotype in this manuscript, homozygotic deletion was found to be embryonic lethal. Would this be the reason that the authors generated a tamoxifen-inducible model? Assuming the authors did not know the reference, it is hard to infer the rationale for generating an inducible knockout model rather than a conventional approach to delete it from the embryo. Was it because the authors knew it was embryonic lethal? If so, it should be clearly mentioned in the introduction part.

Thank you for pointing this out. We have revised the text accordingly to remove any instance of saying ours is the first mouse model for deleting *Immt*. We will include the reference to Feng *et*

al. (PMID: 37107296) to support the rationale for generating an inducible knockout mouse model.

2. The mouse breeding scheme and the genotypes used in the study are not well described in the method section. Did the authors cross the resulting chimera from the ES injection with wild-type first to ultimately obtain the *Immt*-fl/fl by interbreeding? Were then these fl/fl mice crossed with ROSA-CreERT2? If so, what is the exact genotype of control mice as denoted "Immt-WT"? Is it the ROSA-CreERT2 mouse without the loxP cassette? A schematic figure would be greatly helpful for readers to understand as the major claim of this manuscript is the generation of the mouse model.

In regards to the genotype, as whole-body Cre expression itself has a hematologic defective phenotype in mice (PMID: 19380810, 37045870) akin to those the authors observed from the knockout mice in this manuscript, it is critical to include Cre control treated with tamoxifen to attribute the phenotype to the loss of *Immt*.

The reviewer is correct, the verified *Immt*^{flox} pups born following ES injection in C57BL/6J mice were first backcrossed to obtain flox sequences on both alleles. Upon successful generation of the *Immt*^{F/F} genotype, these mice were crossed with WT mice expressing ROSA-CreER^{T2}. The resulting offspring were backcrossed until obtaining the final genotypes of the utilized mice, which were WT, *Immt*^{F/WT}, or *Immt*^{F/F}, all with two copies of ROSA-CreER^{T2}. We have clarified this process in the methods section and included a schematic in Fig 1B.

Regarding the reported hematologic defective phenotype associated with tamoxifen-induced Cre activation in the ROSA-CreER^{T2} mice, we first note that we used a lower dose of tamoxifen (~50 mg/kg) respective to these studies (75–200 mg/kg). The indicated reports cited by the reviewer both indicated that these effects are more prominent on very young (P9–P19) pups (PMID: 37045870 and PMID: 19380810). Indeed, PMID: 37045870 specifically states that in “adult mice the [observed] toxicity is tolerated and animals can quickly recover”. All mice in our experiments were at least 9 weeks old, and thus would not be affected in this way. Furthermore, the effects on bone marrow and circulating immune cells that are reported reverse over time and are not lethal to adults (PMID: 19380810), whereas our bone marrow transplant experiment (using *Immt*^{F/F} BM) is lethal to the recipient WT mice only after tamoxifen treatment to induce *Immt* deletion. Additionally, the H&E and pH3 BM data included WT controls that had at least one copy of ROSA-CreER^{T2}; these data should be reassuring that this effect is specific to loss of *Immt*. Finally, we completed additional experiments using tamoxifen treatment with WT mice that have at least one copy of ROSA-CreER^{T2} to account for any changes in BM that would be associated with Cre expression (Fig 3).

3. The age of mice at the treatment of tamoxifen should be indicated.

Certainly, we have included these important details in our Materials and Methods. The full range of ages was between 2 and 6 months. Importantly, we did not see any variations in deletion efficiency by age or sex.

4. I am surprised that there is no single assay about mitochondria after deleting a mitochondrial gene. It is essential to know whether the in vitro MIC60 KO phenotype is recapitulated in vivo despite the limited scope of the claim in the manuscript. MIC60 loss-of-function results in

mitochondrial fragmentation among many (PMID: 32567732). Because the tamoxifen did not delete the gene in all tissues and did not reduce the MIC60 level to half in mice with hemizygous allele in any of the tissues, the authors at least should check the morphology of mitochondria in the tissues shown in the manuscript and see whether the fragmentation and gene deletion result has a correlation.

We have addressed this point by completing immunohistochemical staining of the mitochondrial marker COX1 from the assessed tissues. These data are now in revised Fig 2E and 3E.

5. The tamoxifen-induced gene editing efficiency in multiple tissues does not match the original paper that the author cited (PMID: 17456738). Thus, the authors should verify the gene deletion efficiency in multiple tissues by orthogonal methods such as PCR assay of exon 3 or southern blot. Also, please include the densitometry analyses of western blot to quantitatively indicate the MIC60 expression levels in different genotypes. Increasing the number of WT should be necessary.

We have completed PCR analysis for the assessed tissues, see revised Fig 2A and Fig 3C. We have also added densitometry analysis for MIC60 expression over the time course of tamoxifen treatment, this is presented in Supplementary Fig S1B and Fig 3D.

6. Please provide the gene deletion efficiency in hematopoietic cells that the authors tested in the manuscript.

We agree, and this corresponds with Reviewer 3's Point #3. We have completed this analysis using PCR of genomic DNA and western blotting from HSPCs (see revised Fig 3C and D).

(Optional) 7. The knockout mouse phenotype apparently does not overlap with any of the human diseases associated with MICOS dysfunction (PMID: 31825482), let alone the neurological phenotypes that the authors mentioned in the discussion presumably due to the incomplete penetrance of tamoxifen to the brain. Could the authors elaborate some thoughts on the discrepancy between the observed phenotype in this manuscript and the human diseases?

A similar point was mentioned by Reviewer 1's Point #2, and we have expanded on our Discussion to include this elaboration as follows:

“There are reports of *Immt* mutations in relation to human diseases. Heterozygous, missense mutations within the mitochondrial targeting sequence of *IMMT* were identified in a population of patients with either sporadic or familial Parkinson's disease (27). When these mitochondrial targeting sequence mutations were expressed in *Drosophila melanogaster*, MIC60 did not properly localize to mitochondria, and this was associated with altered mitochondrial cristae morphology, impaired larva crawling ability, and a significantly reduced lifespan. These results also demonstrate the importance of functional MIC60 in a way that relates back to *IMMT* mutations found in patients with Parkinson's disease. In addition, a recent case study identified two consanguineous cousins presenting with developmental encephalopathy that was likely caused by a homozygous mutation in *IMMT* (31). Indeed, defects in mitochondrial function commonly manifest in relation to neurodegenerative disorders (28, 29, 30). Based on these

studies, one might expect to see neurological symptoms in our tamoxifen-treated *Immt*^{F/F} mice, were it not the case that the intestinal defects are too severe for the mice to survive. Additionally, the Rosa-Cre ERT² used in this model does not express in the brain (19), meaning *Immt* would not be well deleted in this organ following tamoxifen treatment. Despite this limitation, this flox-*Immt* mouse could easily be crossed with other inducible Cre models, such as with Synapsin 1-Cre (32), for future investigations of MIC60 function in the brain, highlighting the potential this mouse model has to expand the field of MICOS and mitochondrial function.”

Reviewer #3 (Comments to the Authors (Required)):

This study seeks to define the consequences of disrupting of a core MICOS subunit, MIC60 (gene name *Immt*), in an in vivo mouse model. The authors generated a global inducible *Immt* knockout and find that, strikingly, the animals die acutely in just over a week. They define two tissues demonstrating pathology that may contribute to this lethality: the intestine, which shows pathophysiology consistent with paralytic ileus, and the bone marrow, which shows hypocellularity across multiple cell types. For the most part, the experimental claims are supported by the data presented. However, there are some claims that, if explored in more detail, would substantially strengthen the manuscript. Additionally, I have a couple of technical concerns that should be addressed to more fully characterize the MIC60 inducible model. This is of particular importance as the authors deem this mouse model "a novel model system for future research." Finally, I believe this manuscript could be significantly strengthened by the inclusion of all experimental data within the main text (rather than having an extended figure), additional citations, and the inclusion of a bona fide discussion section in which they place their findings into a broader context. As it stands currently, the manuscript does not adequately integrate the current state of the MICOS literature, nor does it summarize the novelty of the system and findings to the reader.

Major revisions

1. The authors do a nice job of mapping MIC60 expression across tissues (Figure 1A) and confirming its loss upon tamoxifen treatment across many tissues of interest (Figure 1D, EV1). However, much of this data is included in EV1, and its inclusion in the main text would be helpful for the reader to more fully understand the system and evaluate the conclusions of the paper. For instance, it may be surprising that intestinal KO of *Immt* is what drives lethality in a global KO model, but upon examination of the data, this seems to be true because the knockout of MIC60 occurs in the intestine before (or more completely) than in other tissues such as the kidney and heart. Showing these data in the main text will help the reader to draw these conclusions, which are important for fully understanding the context of the study as well as the model.

In completing a tamoxifen time course in *Immt*^{F/F} mice, we show in revised Fig 2A that we obtain complete loss of the flox-*Immt* alleles in the small intestine, but not in other tissues (colon and spleen). Western blotting over this time course shows reduced MIC60 expression over time, and small intestine appears to be more efficiently reduced at day 3 of treatment relative to the colon or spleen. Though we kept some western blot data in Supplementary Fig S1 (liver, kidney,

heart, and skeletal muscle), we think the data in Fig 2 addresses the Reviewer's main point of there being a difference in deletion efficiency across multiple tissue types after *Immt* deletion.

2. The half-life data shown in Figure 1C is interesting, but it is overstated and somewhat irrelevant to the claims made in the paper. First, stating that MIC60 is a "long-lived protein in MEFs" because it does not decrease in abundance over 24 hours relative to MCL-1 is an overstatement. A recent study looking at isotopic labeling in HeLa cells found MCL-1 to be the shortest-lived mitochondrial protein quantified with a half-life of 45 minutes (PMID: 34847359, Figure S2B). Furthermore, a recent study defining short lived proteins across 4 cultured cell lines found that fewer than 10% of proteins are 'short-lived' with a half-life under 8 hours following translational arrest (PMID: 34626566). Thus, the statement that MIC60 is 'long-lived', based on a cycloheximide time course over 24 hours is an overstatement; it has a longer half-life than MCL-1, but this is not surprising given these datapoints. Nonetheless, the experiment itself is flawed because mitochondrial proteins will almost certainly have different half lives in cultured cells versus in vivo. However, this can be addressed by the inducible nature of the *Immt* knockout mouse model profiled in the paper. The authors should use this mouse to map the half-life of MIC60 across the intestine and the colon to understand the protein turnover kinetics in these tissues. This will not only be valuable to understand MIC60 biology but would also be important for understanding the kinetics and onset of pathology relative to loss of the protein.

We agree and have revised the text accordingly. The cycloheximide treatment in MEFs has been removed from the manuscript, and we now present both genotyping and western blotting data over time, after *Immt* deletion, in multiple tissue types.

3. The protein expression of MIC60 is not shown for any resident cell types in bone marrow, and this should be shown for at least a subset of these cells. However, MIC60 deletion likely does not directly drive pathologies in all hematopoietic cells, such as red blood cells, which are diminished in *Immt* KO mice, but would be predicted to be unaffected by loss of MICOS due to their lack of mitochondria. The fact that the pathology persists in RBCs suggests that hematopoietic stem cells may be affected as well, which could be tested for MIC60 expression after tamoxifen treatment.

The point of assessing MIC60 expression from bone marrow cells was also brought up by Reviewer 2 (see Point #6). We have addressed this through genotyping and western blotting of hematopoietic stem and progenitor cells and included the data in Fig 3.

4. The authors note that tamoxifen is given by oral gavage in the animals, which induces lethality within 9 days. It is notable that intestinal cells would be amongst the first to encounter tamoxifen via this delivery method, which may contribute to the severity of pathophysiology in this cell type. Alternatively, it is possible that MIC60 has a relatively short half-life in the intestine relative to other tissue types, which is consistent with what the authors propose. The authors should test whether tamoxifen administration via a non-oral route, such as intraperitoneal injection, causes lethality on the same timescale and causes intestinal pathology. This need only be done on a small cohort of animals, and the full scope of the study need not be repeated. It should also be noted that if the results of this experiment show different phenotypes than what are detailed in the current figures, it does not invalidate the work, but rather suggests that the

method in which researchers induce *Immt* knockout influences the pathology seen. Given the authors' assertions that this paper is characterizing "a valuable resource" and "a novel model system for future research", this is an important control for understanding the utility of this model, as well as the conclusions that can be drawn from future studies in which it may be used.

We completed intraperitoneal injections with tamoxifen into a cohort of mice, using the same dosage and treatment schedule as for oral gavage. We observed the same intestinal defects (see Fig 1D), and mouse survival in this cohort sharply declined by day 7 post treatment (see Supplementary Fig S1A).

5. The lack of discussion in the manuscript detracts from its potential impact. There is very little context as to why the authors performed the study and what it might mean in the context of the MICOS field. There are no parallels made to human disease, despite the fact that mutations in *IMMT* have been linked to encephalopathy and optic neuropathy (PMID: 34842280). The authors should place their findings into the context of these and other important studies in the MICOS field.

This point was mentioned by Reviewers 1 and 2 as well, and we have expanded on our Discussion to include relevance to human disease and context to the MICOS field, as follows:

“There are reports of *Immt* mutations in relation to human diseases. Heterozygous, missense mutations within the mitochondrial targeting sequence of *IMMT* were identified in a population of patients with either sporadic or familial Parkinson’s disease (27). When these mitochondrial targeting sequence mutations were expressed in *Drosophila melanogaster*, MIC60 did not properly localize to mitochondria, and this was associated with altered mitochondrial cristae morphology, impaired larva crawling ability, and a significantly reduced lifespan. These results also demonstrate the importance of functional MIC60 in a way that relates back to *IMMT* mutations found in patients with Parkinson’s disease. In addition, a recent case study identified two consanguineous cousins presenting with developmental encephalopathy that was likely caused by a homozygous mutation in *IMMT* (31). Indeed, defects in mitochondrial function commonly manifest in relation to neurodegenerative disorders (28, 29, 30). Based on these studies, one might expect to see neurological symptoms in our tamoxifen-treated *Immt*^{F/F} mice, were it not the case that the intestinal defects are too severe for the mice to survive. Additionally, the Rosa-Cre ERT² used in this model does not express in the brain (19), meaning *Immt* would not be well deleted in this organ following tamoxifen treatment. Despite this limitation, this flox-*Immt* mouse could easily be crossed with other inducible Cre models, such as with Synapsin 1-Cre (32), for future investigations of MIC60 function in the brain, highlighting the potential this mouse model has to expand the field of MICOS and mitochondrial function.”

Minor revisions:

1. "Inducible" and "induces" are both used in the title, which is redundant.

We have corrected the title accordingly.

2. The histology in this paper would be more powerful if quantified and analyzed statistically.

We revised the panel with the IHC staining for phosphorylated Histone 3 to include the quantification of positive cells (now presented in Figure 3A). After discussing with our Institutional Comparative Pathology Department, we have quantified the histology by the number of mice in each group that exhibited the described phenotypes, as follows:

“Histological analysis of the intestinal tract (small intestines, colon, and cecum) revealed apoptotic cells, vacuolar degeneration of small intestine enterocytes, dilated intestinal crypts containing bacteria, and submucosal inflammation in the *Immt*^{F/F} mice (observed in 5 out of 5 mice), but not in WT mice (not observed in any of four mice, **Fig 1E**).”

3. The authors find loss of MIC60 in the spleen - are there signs of splenomegaly?

We did not observe splenomegaly – in fact, in measuring the weight of the spleens following tamoxifen treatment by oral gavage, the spleens were slightly smaller in *Immt*^{F/F} relative to WT.

4. The authors claim that "To date, the in vivo impact of MIC60 loss has not been documented." However, a recent study generated Mitofilin (i.e., *Immt*) global knockout mice and this study should be cited. (PMID: 37107296).

Our thanks to the reviewer; this was also brought to our attention by Reviewer 1 (Point #1). We have revised the text accordingly to remove any instance of saying ours is the first mouse model for deleting *Immt*. Furthermore, we included the reference to Feng *et al.* 2023 (PMID: 37107296) to support the rationale for generating an inducible knockout mouse model.

January 23, 2024

Re: Life Science Alliance manuscript #LSA-2023-02329-TR

Dr. Joseph Thomas Opferman
St. Jude Children's Research Hospital
Cell and Molecular Biology
262 Danny Thomas Place
MS 340
Memphis, TN 38105

Dear Dr. Opferman,

Thank you for submitting your revised manuscript entitled "Genetic ablation of Immt induces a lethal disruption of the MICOS complex" to Life Science Alliance. The manuscript has been seen by the original reviewers whose comments are appended below. While the reviewers continue to be overall positive about the work in terms of its suitability for Life Science Alliance, some important issues remain.

Our general policy is that papers are considered through only one revision cycle; however, we are open to one additional short round of revision. Please note that I will expect to make a final decision without additional reviewer input upon re-submission.

Please submit the final revision within one month, along with a letter that includes a point by point response to the remaining reviewer comments.

To upload the revised version of your manuscript, please log in to your account: <https://lsa.msubmit.net/cgi-bin/main.plex>
You will be guided to complete the submission of your revised manuscript and to fill in all necessary information.

B. MANUSCRIPT ORGANIZATION AND FORMATTING:

Sincerely,

Reviewer #1 (Comments to the Authors (Required)):

In the revised manuscript, Rockfield et al have improved their manuscript by including additional control data to look at the

kinetics of their inducible MIC60 knockout in different tissues. They also made several textual revisions that improve the precision of their conclusions. Notably, however, the authors removed claims of studying half-life of MIC60 in different tissues, which was one of the key findings of the original manuscript, presumably due to the aforementioned kinetic data.

In the revision, the authors now claim to study mitochondrial morphology via IHC. However, these conclusions are overstated as this assay is not appropriate to assess mitochondrial morphology. The claim that mitochondria are enlarged and swollen is not supported by the images shown. A shortcoming of the manuscript remains that mitochondrial defects are not studied in this knockout system, which is an important validation of the system as a tool that should be included in order to recommend publication.

As a minor point, references to Figure S2 in the manuscript presumably refer to Figure S1.

Reviewer #2 (Comments to the Authors (Required)):

1. In the initial manuscript, there were MIC60 immunoblots from the hemizygous deletion mice, showing no change. In this revised manuscript, those data were excluded without mentioning. The authors can't make any claim (e.g. lines 103, 118) about the hemizygous mice without showing MIC60 level in those mice. Hemizygous deletion does not necessarily lead to half reduction of the target protein as evidenced by a recent report (PMID: 35338200). Therefore, if this is the case, it should be mentioned in the main text with data. Otherwise, the authors should delete all the claim about hemizygous mice, unless they are presented.

2. I do not see decreases in SAM50 from Figure 2B, 2C, 2D. This is contrary to HSPC in Figure 3D. As authors mentioned the blots are representatives, clearer blots should be presented to represent the reduction of SAM50. If the authors do not see a clear reduction, please mention this in the main text.

3. I cannot tell the difference in mitochondrial morphology assessed by IHC in Figure 2E or Suppl Fig 1E. Given that mitochondrial width is around 200 nm, it is not feasible to tell mitochondrial morphology by the resolution from IHC. What I see instead is that uneven staining pattern of COX1 with disrupted tissue morphology in Immt KO mice, implying dysregulated COX1 activity in the affected tissue. Thus, any morphological claim about mitochondria is not valid. I suggest performing electron microscopy or protein import assay in the isolated mitochondria from the tissue for the validation of mitochondrial dysfunction caused by the disrupted MICOS complex.

Reviewer #3 (Comments to the Authors (Required)):

This study seeks to define the consequences of disrupting of a core MICOS subunit, MIC60 (gene name Immt), in an in vivo mouse model. In the revised manuscript, the authors add multiple important experiments, including a tamoxifen time course in Immt flox/flox mice across multiple tissues, examination of Immt in hematopoietic stem cells, and inclusion of data testing the effects of tamoxifen administered by intraperitoneal injection. These data strengthen the conclusions of the manuscript and bring clarity to the origins of the lethal pathophysiology.

Unfortunately, there are still some concerns with the conclusions drawn by the authors, particularly in the assertions made about mitochondrial morphology in the newly included immunohistochemistry experiments (Figures 2E and 3E). The images displayed do not show any clear mitochondrial morphology as they lack the resolution required to visualize individual organelles. Thus, the conclusions drawn by the authors that "mitochondria appeared more enlarged and swollen in the small intestine, cecum, and colon, but were unchanged in the liver or kidney" (lines 135-6) are unsubstantiated. To make such claims, the authors could use alternative experimental approaches such as fluorescence-based techniques in tissue slices or perhaps electron microscopy. Alternatively, the authors could culture primary cells from their mouse models, although this may not fully address the requests by Reviewer #2 to examine morphology in the tissues shown in the manuscript. As it stands, the conclusions drawn by the authors are neither supported nor refuted by the data due to their lack of interpretability. This should be addressed before this manuscript is accepted for publication.

Three other, less critical points for the authors to consider are:

1. On lines 118-9, the authors note that the heterozygous mice responded similarly to tamoxifen as wild type mice, but indicate that "results not shown". If the authors performed these experiments, they should be included in the supplement for others to evaluate.

2. Supplementary Figure 1A is difficult to interpret as no N is reported. How many mice were tested for viability upon IP tamoxifen administration in this graph? Did every mouse given IP injections of tamoxifen die at day 7?

3. The authors did a nice job of adding material to their discussion and addressing reviewer comments about the potential pathophysiological implications of human IMMT mutations. However, in the final paragraph of the discussion, the authors (rightfully) discuss that their mouse model is inadequate to study some of these pathologies, such as encephalopathy, due to the premature lethality in the global inducible model. This is at odds with the framing of the introduction, which states that "we

present a conditional mouse model that will be beneficial for future, more targeted, research into Immt function and regulation." It seems that the current model is actually quite limited due to the severity of the intestinal defects described in this study. This is the authors' manuscript, and they are free to write it as they wish, but I found the introduction and discussion to be somewhat at odds in terms of communicating the overall goal of the study and the utility of the model.

1 All three reviewers (Reviewer 1's point 2, Reviewer 2's point 3, and Reviewer 3's major point)
2 commented that we would require electron microscopy or other modalities to assess
3 mitochondrial morphology.

4 As requested, we have completed high-resolution scanning transmission electron microscopy
5 (STEM) imaging to assess mitochondrial morphology in tissues in which *Immt* was efficiently
6 deleted (i.e., the small intestine and colon tissues). In these tissues, we observed increased
7 mitochondria size (by multiple parameters) as well as aberrant cristae morphology in *Immt*-
8 deleted mice when compared to WT controls. These new data are presented in **Fig 2D and**
9 **Supplementary Fig S2B-C**.

10 In addition, Reviewer 2 (point 1) and Reviewer 3 (minor point 1) both commented on the lack of
11 data regarding heterozygous *Immt* mice.

12 While our prior submission did include some hemizygous deletion data (e.g., the survival curve
13 presented in **Fig 1C** and the kidney, heart, and muscle tissues western blots presented in the old
14 **Supplementary Fig 1D**), we have now expanded this supplementary figure panel to include the
15 western blots of colon and spleen tissues. These results, particularly from the colon, spleen, and
16 liver samples, demonstrate that there was not a marked change in MIC60 expression in the
17 *Immt*^{F/WT} mice relative to WT controls. We also include representative images of the intestinal
18 tract of *Immt*^{F/WT} mice at day 7 of tamoxifen treatment (see new **Supplementary Fig S1B**),
19 which demonstrates the lack of intestinal defects observed at the same timepoints compared to
20 *Immt*^{F/F} mice. These results are described in the text at lines 94, 97, and 113. Finally, we have
21 ensured we do not have any references to "data not shown" in the manuscript text.

22 Reviewer #1 (Comments to the Authors (Required))

23 In the revised manuscript, Rockfield et al have improved their manuscript by including
24 additional control data to look at the kinetics of their inducible MIC60 knockout in different
25 tissues. They also made several textual revisions that improve the precision of their conclusions.
26 Notably, however, the authors removed claims of studying half-life of MIC60 in different tissues,
27 which was one of the key findings of the original manuscript, presumably due to the
28 aforementioned kinetic data.

29 We have revised the manuscript text in the section titled "*Tamoxifen treatment efficiently*
30 *deletes Immt in the small intestines and affects mitochondrial morphology*" (beginning line
31 109) to emphasize our assessment of MIC60's half-life induced by genetic loss in tissues
32 (presented in revised **Fig 2A-B** and the revised **Supplementary Fig S1D**).

33 In the revision, the authors now claim to study mitochondrial morphology via IHC. However,
34 these conclusions are overstated as this assay is not appropriate to assess mitochondrial
35 morphology. The claim that mitochondria are enlarged and swollen is not supported by the
36 images shown. A shortcoming of the manuscript remains that mitochondrial defects are not
37 studied in this knockout system, which is an important validation of the system as a tool that
38 should be included in order to recommend publication.

39 Please see our response above.

40 As a minor point, references to Figure S2 in the manuscript presumably refer to Figure S1.

41 Our thanks, we have ensured that any references to the old Fig S2 have been removed.

42 Reviewer #2 (Comments to the Authors (Required)):

43 1. In the initial manuscript, there were MIC60 immunoblots from the hemizygous deletion mice,
44 showing no change. In this revised manuscript, those data were excluded without mentioning.
45 The authors can't make any claim (e.g. lines 103, 118) about the hemizygous mice without
46 showing MIC60 level in those mice. Hemizygous deletion does not necessarily lead to half
47 reduction of the target protein as evidenced by a recent report (PMID: 35338200). Therefore, if
48 this is the case, it should be mentioned in the main text with data. Otherwise, the authors should
49 delete all the claim about hemizygous mice, unless they are presented.

50 Please see our response above.

51 2. I do not see decreases in SAM50 from Figure 2B, 2C, 2D. This is contrary to HSPC in Figure
52 3D. As authors mentioned the blots are representatives, clearer blots should be presented to
53 represent the reduction of SAM50. If the authors do not see a clear reduction, please mention this
54 in the main text.

55 We have revised the text (line 126) to indicate that “SAM50 expression was not clearly reduced
56 across the assessed tissues”.

57 3. I cannot tell the difference in mitochondrial morphology assessed by IHC in Figure 2E or
58 Suppl Fig 1E. Given that mitochondrial width is around 200 nm, it is not feasible to tell
59 mitochondrial morphology by the resolution from IHC. What I see instead is that uneven staining
60 pattern of COX1 with disrupted tissue morphology in Immt KO mice, implying dysregulated
61 COX1 activity in the affected tissue. Thus, any morphological claim about mitochondria is not
62 valid. I suggest performing electron microscopy or protein import assay in the isolated
63 mitochondria from the tissue for the validation of mitochondrial dysfunction caused by the
64 disrupted MICOS complex.

65 Please see our response above.

66 Reviewer #3 (Comments to the Authors (Required))

67 This study seeks to define the consequences of disrupting of a core MICOS subunit, MIC60
68 (gene name Immt), in an in vivo mouse model. In the revised manuscript, the authors add
69 multiple important experiments, including a tamoxifen time course in Immt flox/flox mice across
70 multiple tissues, examination of Immt in hematopoietic stem cells, and inclusion of data testing
71 the effects of tamoxifen administered by intraperitoneal injection. These data strengthen the
72 conclusions of the manuscript and bring clarity to the origins of the lethal pathophysiology.

73 Unfortunately, there are still some concerns with the conclusions drawn by the authors,
74 particularly in the assertions made about mitochondrial morphology in the newly included
75 immunohistochemistry experiments (Figures 2E and 3E). The images displayed do not show any
76 clear mitochondrial morphology as they lack the resolution required to visualize individual

77 organelles. Thus, the conclusions drawn by the authors that "mitochondria appeared more
78 enlarged and swollen in the small intestine, cecum, and colon, but were unchanged in the liver or
79 kidney" (lines 135-6) are unsubstantiated. To make such claims, the authors could use alternative
80 experimental approaches such as fluorescence-based techniques in tissue slices or perhaps
81 electron microscopy. Alternatively, the authors could culture primary cells from their mouse
82 models, although this may not fully address the requests by Reviewer #2 to examine morphology
83 in the tissues shown in the manuscript. As it stands, the conclusions drawn by the authors are
84 neither supported nor refuted by the data due to their lack of interpretability. This should be
85 addressed before this manuscript is accepted for publication.

86 Please see our response above.

87 Three other, less critical points for the authors to consider are:

88 1. On lines 118-9, the authors note that the heterozygous mice responded similarly to tamoxifen
89 as wild type mice, but indicate that "results not shown". If the authors performed these
90 experiments, they should be included in the supplement for others to evaluate.

91 Please see our response above. Importantly, we have ensured we do not have any reference to
92 "data not shown" in the manuscript text; all data is presented.

93 2. Supplementary Figure 1A is difficult to interpret as no N is reported. How many mice were
94 tested for viability upon IP tamoxifen administration in this graph? Did every mouse given IP
95 injections of tamoxifen die at day 7?

96 We have revised this survival curve – notably, no WT mouse ever fell ill, and all *Immt*^{F/F} mice
97 were moribund on day 7 of tamoxifen treatment by intraperitoneal injection. In our revised
98 graph, we include all 5 WT mice, 5 *Immt*^{F/WT} mice, and 5 *Immt*^{F/F} mice, and have made sure these
99 details are included in the figure legend.

100 3. The authors did a nice job of adding material to their discussion and addressing reviewer
101 comments about the potential pathophysiological implications of human IMMT mutations.
102 However, in the final paragraph of the discussion, the authors (rightfully) discuss that their
103 mouse model is inadequate to study some of these pathologies, such as encephalopathy, due to
104 the premature lethality in the global inducible model. This is at odds with the framing of the
105 introduction, which states that "we present a conditional mouse model that will be beneficial for
106 future, more targeted, research into Immt function and regulation." It seems that the current
107 model is actually quite limited due to the severity of the intestinal defects described in this study.
108 This is the authors' manuscript, and they are free to write it as they wish, but I found the
109 introduction and discussion to be somewhat at odds in terms of communicating the overall goal
110 of the study and the utility of the model.

111 A primary reason we would not be able to use this model as is to study human neurological
112 pathologies is because ROSA-CreER^{T2} does not express in the brain (see PMID: 17456738).
113 While the intestinal defects are severe when tamoxifen is provided to the ROSA-CreER^{T2} mice,
114 Cre expression could still be induced using tissue-specific promoters following administration of

115 a Cre-expressing adeno-associated virus, or by crossing the flox-*Immt* mice to mice expressing a
116 tissue specific Cre, as we referenced in our discussion (line 224):

117 “Indeed, defects in mitochondrial function commonly manifest in relation to
118 neurodegenerative disorders (29-31). Based on these studies, one might expect to see
119 neurological symptoms in our tamoxifen-treated ROSA-CreER^{T2} *Immt*^{F/F} mice. However, the
120 ROSA-CreER^{T2} used in this model does not express in the brain (19), meaning *Immt* would
121 not be deleted in this organ following tamoxifen treatment. Despite this limitation, this flox-
122 *Immt* mouse could easily be crossed with other inducible Cre models, such as with Synapsin
123 1-Cre (32), for future investigations of MIC60 function in the brain, highlighting the
124 potential this mouse model has to expand the field of MICOS and mitochondrial function.”

February 27, 2024

RE: Life Science Alliance Manuscript #LSA-2023-02329-TRR

Dr. Joseph Thomas Opferman
St. Jude Children's Research Hospital
Cell and Molecular Biology
262 Danny Thomas Place
MS 340
Memphis, TN 38105

Dear Dr. Opferman,

Thank you for submitting your revised manuscript entitled "Genetic ablation of Immt induces a lethal disruption of the MICOS complex". We would be happy to publish your paper in Life Science Alliance pending final revisions necessary to meet our formatting guidelines.

- please be sure that the authorship listing and order is correct
- please add the Twitter handle of your host institute/organization as well as your own or/and one of the authors in our system

A. FINAL FILES:

B. MANUSCRIPT ORGANIZATION AND FORMATTING:

****It is Life Science Alliance policy that if requested, original data images must be made available to the editors. Failure to provide**

original images upon request will result in unavoidable delays in publication. Please ensure that you have access to all original data images prior to final submission.**

The license to publish form must be signed before your manuscript can be sent to production. A link to the electronic license to publish form will be available to the corresponding author only. Please take a moment to check your funder requirements.

Sincerely,

March 1, 2024

RE: Life Science Alliance Manuscript #LSA-2023-02329-TRRR

Dr. Joseph Thomas Opferman
St. Jude Children's Research Hospital
Cell and Molecular Biology
262 Danny Thomas Place
MS 340
Memphis, TN 38105

Dear Dr. Opferman,

Thank you for submitting your Research Article entitled "Genetic ablation of Immt induces a lethal disruption of the MICOS complex". It is a pleasure to let you know that your manuscript is now accepted for publication in Life Science Alliance. Congratulations on this interesting work.

DISTRIBUTION OF MATERIALS:

Again, congratulations on a very nice paper. I hope you found the review process to be constructive and are pleased with how the manuscript was handled editorially. We look forward to future exciting submissions from your lab.

Sincerely,
